# Screen-Printed Flexible Thermoelectric Device Based on Hybrid Silver Selenide/PVP Composite Films

**DOI:** 10.3390/nano11082042

**Published:** 2021-08-11

**Authors:** Dan Liu, Yaxin Zhao, Zhuqing Yan, Zhidong Zhang, Yanjun Zhang, Peng Shi, Chenyang Xue

**Affiliations:** 1Key Laboratory of Instrumentation Science and Dynamic Measurement, Ministry of Education, North University of China, Taiyuan 030051, China; zhaoyaxinnuc@163.com (Y.Z.); yan19834532102@163.com (Z.Y.); zdzhang@nuc.edu.cn (Z.Z.); zhangyanjun2000@163.com (Y.Z.); 2Electronic Materials Research Laboratory, Key Laboratory of the Ministry of Education and International Center for Dielectric Research, School of Electronic and Information Engineering, Xi’an Jiaotong University, Xi’an 710049, China; spxjy@mail.xjtu.edu.cn

**Keywords:** Ag_2_Se, polyvinyl pyrrolidone, screen printing, composite films, thermoelectric generator

## Abstract

In recent years, the preparation of flexible thermoelectric generators by screen printing has attracted wide attention due to easy processing and high-volume production. In this work, we propose an n-type Ag_2_Se/polymer polyvinylpyrrolidone (PVP) film based on screen printing and investigate the effect of PVP on thermoelectric performance by varying the ratio of PVP. When the content ratio of Ag_2_Se to PVP is 30:1, i.e., PI30, the fabricated PI30 film has the best thermoelectric property. The maximum power factor (PF) of the PI30 is 4.3 μW·m^−1^·K^−2^, and conductivity reaches 81% of its initial value at 1500 bending cycles. Then, the film thermoelectric generator (F-TEG) fabricated by PI30 is tested for practical application; the output voltage and the maximum output power are 21.6 mV and 233.3 nW at the temperature difference of 40 K, respectively. This work demonstrates that the use of PVP combined with screen printing to prepare F-TEG is a simple and rapid method, which provides an efficient preparation solution for the development of environmentally friendly and wearable flexible thermoelectric devices.

## 1. Introduction

Over the past decade, wearable consumer electronic devices such as wireless headphones, smart glasses, and smart watches have become increasingly popular in human life. Although these devices have gradually decreased power consumption, they still require battery power or charging after prolonged use [1,2,3]. Among many potential power generation methods, thermoelectric generators (TEGs) are an ideal energy supply device. They can directly convert the temperature difference (ΔT) between the human body temperature and the environment into electrical energy and require no maintenance [4,5]. The conversion efficiency of TEGs is mainly determined by thermoelectric materials. The key parameter describing the thermoelectric performance of materials is the figure of merit ZT = *S*^2^*σT/κ*, where *S*, *σ*, *κ* and *T* are Seebeck coefficients, electrical conductivity, thermal conductivity, and thermodynamic temperature. To achieve high ZT, TE materials should have a high power factor (PF = *S*^2^*σ*) and low *κ* [6,7,8,9].

The traditional bulk TEG manufacturing process is expensive and has material wastage, which typically involves many processes such as powder hot pressing, polishing, cutting, assembling, and joining the thermoelectric elements [10,11]. In contrast, the film thermoelectric generators (F-TEGs) not only eliminate expensive processing steps and can be fabricated to the desired size and shape, but also reduce material waste by eliminating the additional cutting to fabricate TE legs. F-TEGs can be generally prepared by sputtering [12,13], electrochemical deposition [14], inkjet printing [15,16], and screen printing [17,18,19,20,21]. All of these methods are readily available for microfabrication, which provides opportunities for TEGs as a micro-dynamic source for electronics. Among them, screen printing is an efficient and low-cost manufacturing technology with low processing temperature. For instance, Ju et al. fabricated the Bi_2_Te_3_ and Sb_2_Te_3_ flexible TEG by screen printing with an output voltage of 85 mV at ΔT of 50 K [17]. Liu et al. similarly used screen printing to prepare an In_2_O_3_/ITO thermocouple that had an output voltage of 53.5 mV at a thermal junction of 1270 °C [20]. Varghese et al. reported a screen-printed flexible and high-performance Bi_0.4_Sb_1.6_Te_3_ p-type film with a voltage of 60 mV at ΔT of 80 K [21]. In general, the complete TEG consists of p-type and n-type thermoelectric material modules, but relatively little research has been completed on n-type materials due to ease of oxidization. Many thermoelectric materials are being explored for power generation applications, such as GeTe [22], PbTe [23], half-Heusler [24], and skutterudites [25]. The well-studied n-type material of Bi_2_Te_3_ is brittle, and the element Te is rare and toxic, which is not favorable for practical applications in wearable devices [17,26,27,28]. As an “electron crystal, phonon liquid” n-type material, silver selenide (Ag_2_Se) is environmentally friendly and abundant, with high electrical conductivity and low thermal conductivity at room temperature. It has become an ideal Bi_2_Te_3_ replacement material and is the most promising n-type material in recent years [29,30,31,32,33]. Mallick et al. initially prepared a flexible-folded TEG consisting of 13 thermocouples using screen printing with an output voltage of 181.4 mV at ΔT of 110 K, where Ag-Se ink was used as n-type leg and PEDOT:PSS as the p-type leg. Later on, Ag-Se ink was applied to 3D printing, three shapes of samples were prepared, and the output voltage was 55 mV at 70 K [29,30,31]. To obtain n-type flexible thermoelectric devices with better performance, some researches have been conducted on composite organic and inorganic materials to make them flexible and to improve the thermoelectric properties of the composites [34,35]. However, most conducting polymers are p-type, and n-type polymers are largely unstable or do not perform well. In contrast, the insulating polymer is more stable and can be applied to n-type TEGs. The insulating polymer polyvinylpyrrolidone (PVP) has excellent film formation and adhesive properties, and its excellent physiological inertness does not cause skin irritation, which allows it to be used as a binding agent in printed films [36,37,38,39,40]. For example, Ankireddy et al. prepared films using PVP-K30 as a binding agent and evaluated the influence of ink composition consisting of carbon, nickel, and silver on thermoelectric properties [37]. Ke et al. used PVP-K90 as the viscosity regulator to obtain water-soluble AgNWs conductive inks with printability, which were directly screen-printed on soft stretchable textiles to obtain good electrical conductivity [39]. For Ag_2_Se, as an inorganic thermoelectric material with good biocompatibility, there are few reports of screen-printing method to prepare Ag_2_Se flexible wearable energy harvesters easily and quickly in large areas. In addition, PVP as an addition of binder can significantly affect the thermoelectric properties of TE films. Therefore, it is necessary to investigate the carrier transport characteristics of Ag_2_Se composite film with PVP to optimize its thermoelectric performance, which will give a promising solution for the preparation of high-performance flexible thermoelectric films by batch screen-printing in the future.

In this work, we propose a simple method to prepare Ag_2_Se/PVP films based on screen printing, and the effect of PVP on thermoelectric performance is investigated by varying the ratio of Ag_2_Se to PVP. Then, the Ag_2_Se/PVP F-TEG is connected to the circuit as a power source, with temperature difference and external load resistance changed to test the output voltage and maximum output power. The thermoelectric materials are manufactured directly into devices by the screen-printing process, which not only eliminates material waste but also reduces the manufacturing cost compared to traditional device fabrication methods.

## 2. Materials and Methods

### 2.1. Materials

Ethanol, ethylene glycol (EG), L-ascorbic acid, β-Cyclodextrin, selenium dioxide (SeO_2_), and silver nitrate (AgNO_3_) were bought from Aladdin Industrial Corporation. Polyvinyl pyrrolidone (PVP-K30, MW = 40,000) was purchased from Yatai Chemical Co., Ltd (Wuxi, China). Terpineol was bought from Xilong Science Co., Ltd. (Shantou, China). All reagents were used directly without purification.

### 2.2. Preparation of Ag_2_Se NRs and Ag_2_Se/PVP Film

All containers and tools were first ultrasonically cleaned. A total of 0.5 g SeO_2_ and 0.5 g β-cyclodextrin was magnetically stirred into 100 mL deionized water. A total of 2 g L-ascorbic acid was magnetically stirred into 100 mL deionized water, then slowly added dropwise to the above-mixed solution. After stirring for about 4 h, the supernatant was separated by centrifugation at 8500 rpm for 5 min and then washed twice using alternating deionized water and ethanol. Then, it was dispersed in ethanol and left for 36 h until the next experiment.

The Se nanorods (NRs) of the above solution were transferred into 10 mL EG and then sonicated dispersion. According to the molar ratio of Ag and Se, a certain amount of AgNO_3_ was magnetically stirred into 10 mL EG, and then slowly added dropwise to the Se NRs solution. Additionally, an amount of L-ascorbic acid was added into 20 mL deionized water and then slowly drop in the above-mixed dispersion. The molar ratio of L-ascorbic acid:AgNO_3_ was 3:1. After continuous stirring for 4 h to completely finish the reaction, the precipitates were collected by centrifugation at 8500 rpm for 5 min and washed twice with alternating deionized water and ethanol. Finally, the Ag_2_Se NRs were obtained by drying at 60 °C under nitrogen atmosphere.

A certain amount of PVP was dispersed and dissolved in 0.4 g terpineol, and 0.5 g Ag_2_Se was dispersed in the above solution under sufficient stirring. The polyimide (PI) substrate was ultrasonically cleaned to ensure its surface was free of impurities. After the above mixture was screen-printed on PI substrate with a 200 mesh screen, it was heated and cured at a constant temperature for 10 min to remove the terpineol in a nitrogen environment. The screen printing and sintering process was repeated three times until the mixture was used up. The effect of PVP content on the performance of films was investigated by adjusting the PVP content to determine the content ratio of Ag_2_Se to PVP as 10:1, 20:1, and 30:1, and naming them PI10, PI20, and PI30, respectively. The films consisted of 6 strips, each with a length of 40 mm × 5 mm and a spacing of 10 mm. The F-TEGs used conductive silver adhesive to connect the legs to reduce additional resistance. The manufacturing process and assisted sintering of the Ag_2_Se/PVP film using low-cost screen printing are shown in Figure 1.

### 2.3. Measurement of Ag_2_Se NRs and F-TEG

The phase composition of the Ag_2_Se NRs was determined by X-ray diffraction (XRD, DX-2700, Dandong, China). The surface morphology and thicknesses of these composite films were observed by the field emission scanning electron microscope (FESEM, JSM-7001F, Tokyo, Japan). Simultaneously, the energy spectrum analysis was measured by X-ray energy dispersive spectroscopy (EDS). The Seebeck coefficient (*S*) and electrical resistivity (*ρ*) were measured by the standard four-probe method in helium atmosphere (Linseis, LSR-3, Selb, Germany), and the measurement error of *S* and *ρ* was about ±5%. The carrier concentration (*n*) and mobility (*μ*) were measured by the Van der Pauw method in the nitrogen atmosphere (Linseis, HCS, Selb, Germany).

A test circuit was set up with the F-TEG as the power supply for measuring its output performance. The hot side of the F-TEG obtained a variable high temperature through a heating stage, while the cold side maintained a stable room temperature through a circulating water cooling device. The output voltage could be measured by changing the heating temperature. The temperature was measured by a non-contact infrared thermometer.

## 3. Results and Discussion

The X-ray diffraction (XRD) of the prepared Ag_2_Se NRs is shown in Figure 2A. All diffraction peaks can be indexed to Ag_2_Se (PDF#24-1041), which indicates Ag_2_Se NRs produced no significant impurities. Additionally, it appears as the orthogonal phase β-Ag_2_Se, which is the space point group P2_1_2_1_2_1_. The FESEM images of the surfaces of PI10, PI20, and PI30 are shown in Figure 2B–D. The lengths of the Ag_2_Se NRs are greater than 4.5 μm, and the diameters are about 250 nm, which are nanorods. It can be seen that the addition with different contents of PVP did not change the surface of these films because PVP is an insulating polymer and does not react with Ag_2_Se NRs. The thicknesses of these films are shown in Appendix A. The thicknesses are in the range of 100–110 μm, and the measured thickness deviation is within 5 μm, which demonstrates the uniformity of screen printing. The mapping of the elements C, O, S, Ag, and Se is shown in Figure 2E. Additionally, the details of energy dispersive spectrum (EDS) analysis are shown in Appendix A. For PVP, it is mainly composed of C, O, and N elements. The characteristic spectrum line of the O element is very close to that of the N element, which makes the detected element O, while for S, its content is minimal, which may come from EDS detection error. As the major functional role of thermoelectric materials in the composite film, the analysis of Ag_2_Se is the top priority. The element ratio analysis of the surface shows that the ratio of Ag to Se is approximately 2:1, which agrees with the stoichiometric ratio. Additionally, the Ag_2_Se NRs are uniformly distributed in the composite films. Although there is a large error in calculating the element’s content by EDS, especially for the element with a lighter atomic mass, the result reveals that the Ag_2_Se and PVP are mixed well in the composite film. In short, the uniform composite film has been prepared well by the screen-printing method and can be further used to fabricated a flexible TE generator.

As shown in Figure 3, the Seebeck coefficient (*S*), electrical conductivity (*σ*), and power factor (PF) of PI10, PI20, and PI30 films were tested to study the thermoelectric properties from 300 K to 410 K. In Figure 3A, the negative value of the Seebeck coefficient indicates that Ag_2_Se is an n-type TE material. These Seebeck coefficients of films also exhibit a similar tendency of changes, showing that the addition of PVP does not affect the intrinsic conductivity type of the Ag_2_Se. Additionally, in the case of PI30, the Seebeck coefficient has a sharp decrease from 58.5 μV·K^−1^ to 36.0 μV·K^−1^ from 390 K to 410 K, which is due to the phase transition of Ag_2_Se from a low-temperature semiconductor phase to a high-temperature superionic conductor phase near 407 K. In Figure 3B, the electrical conductivities of these films maintain a rising trend with temperature due to intrinsic property of semiconductor, such as the PI30 increases from 10.7 S·cm^−1^ at 300 K to 14.9 S·cm^−1^ at 410 K. In addition, the electrical conductivities of these films decrease with the increase of PVP content because of the increase in insulation. After decreasing the PVP content again, the dried films no longer adhered to the PI substrate and had extremely poor flexibility, as shown in Appendix A. Under the simultaneous action of the Seebeck coefficient and the electrical conductivity, PFs (PF = *S*^2^*σ*) of these films have a trend of increase with the decrease of PVP content; as shown in Figure 3C, the PF of PI30 with the least PVP content shows the best TE performance. At the same time, the PF of PI30 has a maximum value of 4.3 μW·m^−1^·K^−2^ at 390 K, while its PF drops suddenly to 1.9 μW·m^−1^·K^−2^ at 410 K, which is due to phase transition of Ag_2_Se.

To explain the variation of the Seebeck coefficient and the electrical conductivity with temperature, the three films were tested for the Hall effect based on the Vanderbilt method. The measured carrier concentration (*n*) and mobility (*μ*) change with temperature from 300 K to 410 K, as shown in Figure 4. Taking PI30 as an example, the carrier concentration is 3.31 × 10^18^ cm^−3^ and the mobility is 17.81 cm^2^·V^−1^·S^−1^ at 300 K. When the temperature rises from 400 K to 410 K, the carrier concentration suddenly increases from 4.28 × 10^18^ cm^−3^ to 6.31 × 10^18^ cm^−3^ and mobility decreases from 18.21 cm^2^·V^−1^·S^−1^ to 14.36 cm^2^·V^−1^·S^−1^. The phase change of Ag_2_Se at 407 K causes changes in the carrier concentration and mobility, which in turn lead to changes in the Seebeck coefficient and electrical conductivity. The internal correlations are as follows [41]:(1)$$S=\frac{8\pi^{2}k_{B}^{2}}{3eh^{2}}m^{*}T{\left(\frac{\pi }{3n}\right)}^{2/3}$$
(2)σ=neμ
where *k_B_* is the Boltzmann constant, *e* is the electron quantity, *h* is the Planck constant, and *m** is the effective carrier mass. Below the phase transition temperature of 407 K, the Ag_2_Se material shows intrinsic excitation with increasing temperature, resulting in a slowly increasing trend of carrier concentration. The results are deduced from Equations (1) and (2): the absolute value of the Seebeck coefficient decreases, and electrical conductivity increases slowly. When the phase transition temperature is reached, since Ag_2_Se transforms from the low-temperature semiconductor phase to the high-temperature superionic conductor phase, the presence of two structures increases the disorder of the system. Compared to the single structure, the scattering effect of carrier concentration is improved, and therefore, the Seebeck coefficient is significantly reduced. At this time, structural transformation and scattering cause a decrease in mobility, which has an impact on electrical conductivity, but the increase in carrier concentration has a greater impact on electrical conductivity; hence, electrical conductivity increases at this point. The change in carrier concentration and mobility leads to an increase in electrical conductivity and a decrease in the Seebeck coefficient, which ultimately affects the TE properties. In addition, although the thermal conductivities of these composite films have not been measured due to the fact that the thermal conductivities of the composite films with a tiny thermal capacitor can be neglected, in practice, the enhancement of the scattering effect will also make the thermal conductivity *k* (*k* = *k_e_* + *k_l_*; *k_e_* is electronic thermal conductivity, and *k_l_* is lattice thermal conductivity) decrease due to the reduction of *k_e_* (*k_e_* = *σLT* = *μneLT*; *L* is Lorenz factor, *μ* is the mobility ratio of carriers, *n* is the carrier concentration). At the same time, PVP, as an organic matter with low thermal conductivity, will make the thermal conductivities of the composite films lower than that of pure Ag_2_Se, which means PI10 will have the lowest thermal conductivity. In a few words, the phenomenon demonstrates that the TE performance of the composite films before the phase transition temperature can be kept stable, which is suitable for the daily use of wearable devices.

As the energy source for wearable devices, the flexibility of TEGs is also significant. Therefore, the three films are tested for flexibility to further verify the adhesion and film-forming properties of PVP, and the results are shown in Figure 5. The films were bent repeatedly around an 8 mm diameter circular rod, and the internal resistance of each module increases with the increase of bending times. After bending 1500 times, the conductivity of PI10 was reduced to 93% of the initial state, PI20 to 86%, and PI30 to 81%, which meant that the module with the higher PVP content had the best flexibility, and therefore, the film with PVP had excellent bonding and film-forming properties. Moreover, PVP has excellent physiological inertness and biocompatibility, which has great potential in the research of wearable flexible devices. In conclusion, screen-printing provides a solution for preparing composites of multiple materials on different substrates, such as fabric, which is more suitable for large volume wearable preparation.

After flexibility testing, the feasibility of F-TEG as a wearable energy source was verified by a series of F-TEG output performance tests (the circuit is shown in Appendix A). By changing the temperature difference between the two ends of F-TEG, the output voltage and output power under a certain temperature difference were tested; the results are shown in Figure 6A,B. There is a linear relationship between temperature difference and voltage in Figure 6A. When the temperature differences are 20 K and 40 K, the output voltages of F-TEG are 11.1 mV and 21.6 mV, and the maximum output powers are 61.1 nW and 233.3 nW, respectively. The voltage *U_l_* is calculated according to the following formula [42]:(3)$$U_{l}=\left(1-\frac{R_{in}}{R_{in}+R_{l}}\right)\times U_{0}$$
where *U*_0_ is the open-circuit voltage of the F-TEG and *R_in_* is the internal resistance of the F-TEG. Therefore, under a certain temperature difference, the output voltage will increase with the increase of the load resistance and eventually become infinitely close to the open-circuit voltage. When the load resistance is equal to the internal resistance of the F-TEG module at about 500 Ω, the maximum output power is obtained at this time. The F-TEG power density *P_d_* is 0.08 W·m^−2^ at a temperature difference of 40 K, which can be calculated as follows:(4)$$P_{d}=\frac{P_{\max}}{N\times S}$$
where *P_d_* is the power density, *P_max_* is the maximum output power, *N* is the number of F-TEG strips, and *S* is the cross-sectional area. In Figure 6C, F-TEG wraps around the arm with one side in direct contact with the skin and the other side isolated from the skin by a bubble wrap. The temperature difference of about 4 K between the arm and the environment generates a voltage of 1.7 mV (the temperature is measured with a non-contact infrared thermometer). In Figure 6D, the same F-TEG is wrapped in a beaker containing hot water, and a voltage of 7.3 mV is generated from a temperature difference of 15 K between the upper (cold) and lower (hot) surfaces. The result indicates that screen-printed Ag_2_Se films have great potential for the application of wearable devices.

## 4. Conclusions

In conclusion, we reported a facile method for the preparation of Ag_2_Se/PVP film based on screen printing with lower processing temperature and a simple handling process. After Ag_2_Se was dispersed in the PVP terpineol solution, it was screen-printed on the PI substrate and then heated to remove the terpineol. The PI30 had the largest PF of 4.3 μW·m^−1^·K^−2^, but its flexibility was the worst. Its conductivity reduced to 81% of the initial value after bending 1500 times. On the contrary, PI10 had the best flexibility but the smallest PF value. This is because PVP reduces conductivity while increasing the flexibility of the film. F-TEG fabricated by PI30 films was connected to the test circuit as a power supply; the output voltage and the maximum output power were 21.6 mV and 233.3 nW at the temperature difference of 40 K, respectively. This work shows that PVP as a binder combined with screen printing to prepare F-TEGs is a simple and fast method that can provide ideas for preparing other material complexes on different substrates.

## Figures and Tables

**Figure 1 nanomaterials-11-02042-f001:**
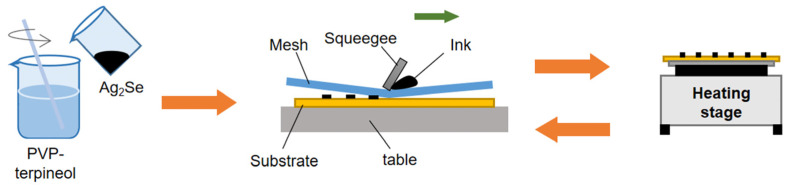
Schematic of the manufacturing process of the Ag_2_Se/PVP film.

**Figure 2 nanomaterials-11-02042-f002:**
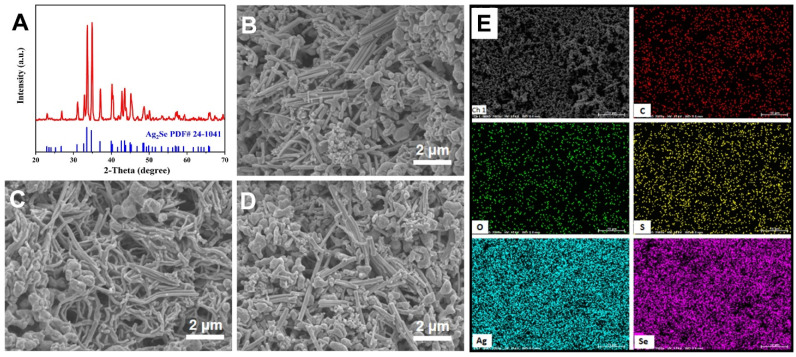
(**A**) The XRD patterns of Ag_2_Se NRs; the FESEM images of (**B**) PI10, (**C**) PI20, and (**D**) PI30; (**E**) the EDS of Ag_2_Se NRs.

**Figure 3 nanomaterials-11-02042-f003:**
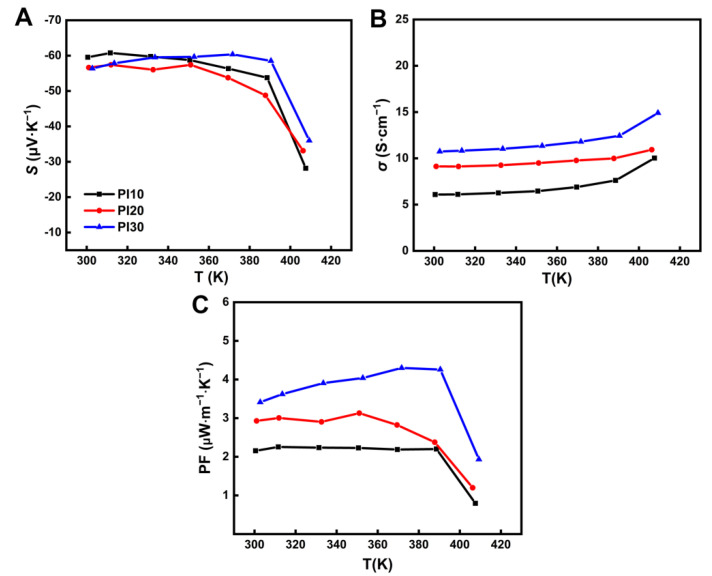
Temperature dependence for (**A**) Seebeck coefficient, (**B**) electrical conductivity, (**C**) power factor of the PI10, PI20, and PI30.

**Figure 4 nanomaterials-11-02042-f004:**
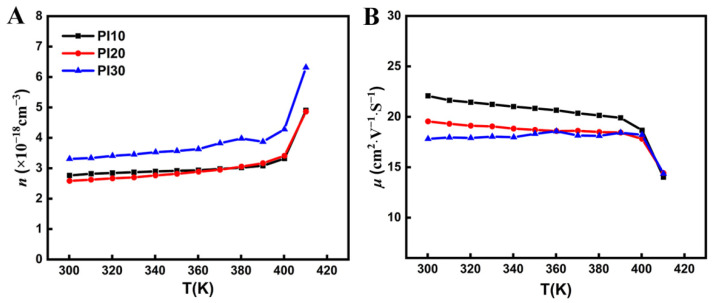
Temperature dependence for (**A**) carrier concentration; (**B**) mobility of PI10, PI20, and PI30.

**Figure 5 nanomaterials-11-02042-f005:**
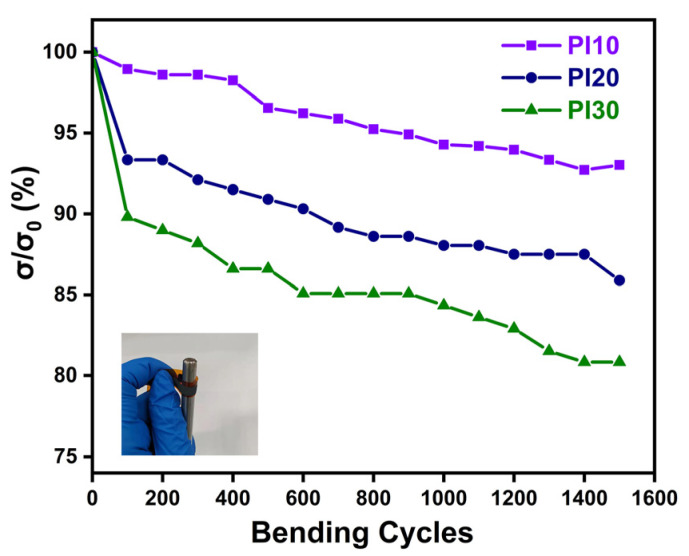
The flexibility of PI10, PI20, and PI30.

**Figure 6 nanomaterials-11-02042-f006:**
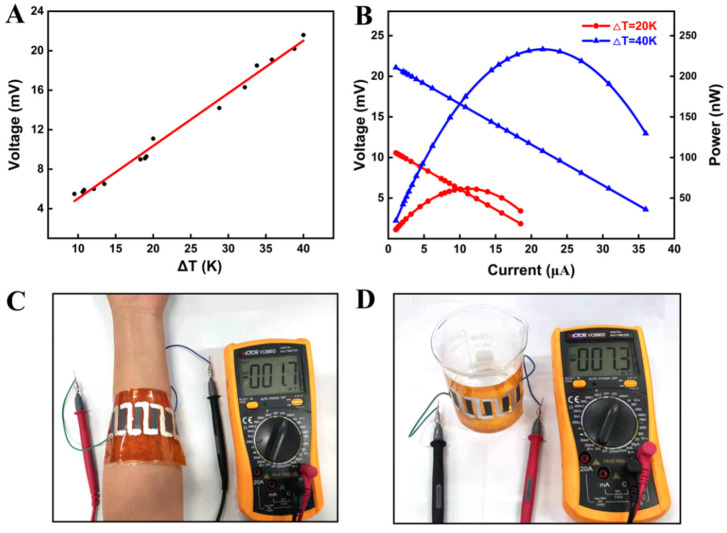
(**A**) PI30 F-TEG open voltage changes with ΔT. (**B**) The relationship between the output voltage and current and power at ΔT = 20 K and 40 K. (**C**) The image of 1.7 mV voltage generated by ΔT between the arm and the environment. (**D**) The image of 7.3 mV voltage generated by ΔT between the upper (cold) and lower (hot) surfaces of the beaker.

## Data Availability

The data are available within the manuscript and the corresponding supporting information file.

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
