# Peer review of "Screen-Printed Flexible Thermoelectric Device Based on Hybrid Silver Selenide/PVP Composite Films"

_nanomaterials, 2021, doi:10.3390/nano11082042_

Round 1

Reviewer 1 Report

It is a great article dealing with screen-printed flexible thermoelectric device based on hybrid silver selenide/PVP composite films.

I have several recommendations for gaining more interest to the paper by a broad range of researchers, dealing with the development of various classes of thermoelectric materials which can be scree printed.

For those, I would recommend to include in the introduction, a sentence in the form of, many thermoelectric materials are being explored for power generation applications, such as GeTe [1],  PbTe [2], half-Heusler [3] and skutterudites [4] with the following references:

  1. Yaniv Gelbstein, Joseph Davidow, Ehud Leshem, Oren Pinshow and Strul Moisa, Significant lattice thermal conductivity reduction following phase separation of the highly efficient GexPb1-xTe thermoelectric alloys, Physica Status Solidi B 251(7) 1431-1437 (2014).
  2. Dana Ben-Ayoun, Yatir Sadia, and Yaniv Gelbstein, High temperature thermoelectric properties evolution of Pb1-xSnxTe based alloys, Journal of Alloys and Compounds 722 33-38 (2017).
  3. Yunfei Xing et al., High-efficiency half-Heusler thermoelectric modules enabled by self-propagating synthesis and topologic structure optimization, Energy Environ. Sci. 12 3390 (2019).
  4. Rull-Bravo et al., Skutterudites as thermoelectric materials: revisited RSC Adv., 5, 41653 (2015).

Following taking into accounts the minor revisions specified above I will be glad to recommend on acceptance of the manuscript.

Reviewer 2 Report

Liu and coworkers have explored the transport properties of screen printed Ag2Se/polyvinylpyrrolidone hybrid thermoelectrics. These devices are flexible up to 1500 bending cycles. The authors have provided a pathway to environmentally friendly and wearable flexible thermoelectric devices, which makes this work interesting. The following issues should be addressed by the authors.

1) The obtained power factor is very low compared to other thermoelectric devices, including flexible Ag/Ag2Se/nylon system (see https://doi.org/10.1021/acsami.1c02194). This needs to be discussed.

2) The peak performance has been obtained with the temperature gradient of 40 K. Is such a gradient reasonable for wearable devices? Please elaborate.

3) Please specify the deposition method from Ref. 14 on page 2. It has only been referred to as “deposition”, which can essentially be related to quite many methods and hence it is uncertain.

4) While the technical motivation for the current manuscript is clear, the scientific part remains weak in the introduction. What physics or chemistry are you after? What’s really unknown? Please improve the motivation of your work.

5) If O originates only from polyvinylpyrrolidone, the amount of O should be equal to N, but the authors have not reported the N content in the samples. Another issue with the composition is that the C to O ratio should be 6, but this is not the case based on EDS. Hence, the measured composition is not reliable.

6) Some acronyms are wrong (e.g. FESEM is “field scanning emission electron microscope” according to the authors on page 3 – obviously “emission” and “scanning” were swapped). There is quite some confusion with the identical or similar acronyms, but with the different meaning (e.g. “polyimide (PI), “When the 14 content ratio of Ag2Se to PVP is 30:1, i.e. PI30”, “insulating polymer (IP)”…). Please carefully check all the acronyms and make the necessary corrections.

7) How can authors make claims on elasticity (“had extremely poor elasticity” on page 5) without any measurement of elasticity? This is a claim without any evidence and hence not acceptable in scientific publications.

8) Eq. (1) and (2) as well as the corresponding discussions are provided without references. Please provide the necessary references.

9) The authors have discussed the scattering of change carriers. Could this be related to the thermal conductivity?

10) The use of italics for variables is not consistent (e.g. mobility in Fig. 4 vs. Eq. (2) and the corresponding text). Please make it consistent throughout for all the variables used.

11) The structure of the manuscript is very strange. Instead of conclusions, the authors have only provided discussions towards the end of the main body of the text. This is not adequate.

12) The authors have claimed “excellent physiological inertness and biocompatibility” on page 6. This is either a claim without any evidence or without a reference. Either way, this is not acceptable.

13) What is the physical origin of the behavior shown in Fig. 5?

14) Please remove Eq. (3). This is too trivial.

Round 2

Reviewer 2 Report

The authors have improved the manuscript to some extent. However, some issues are still open. The authors have admitted that EDS is not a suitable technique for determination of light elements, but they have done nothing about it. Impurities (such as oxygen) can drastically affect the transport properties. Furthermore, discussions on the thermal conductivity should be at least in part included in the manuscript. Similar goes for the discussions on Fig. 5 (the authors should explain the physical origin of the observed behavior). It is also not understandable as to why the authors refuse to rename the section 4 into conclusions. Hence, the revision is not satisfactory.
